# Therapeutic Potential of Natural Products in Treating Neurodegenerative Disorders and Their Future Prospects and Challenges

**DOI:** 10.3390/molecules26175327

**Published:** 2021-09-02

**Authors:** Md. Habibur Rahman, Johny Bajgai, Ailyn Fadriquela, Subham Sharma, Thuy Thi Trinh, Rokeya Akter, Yun Ju Jeong, Seong Hoon Goh, Cheol-Su Kim, Kyu-Jae Lee

**Affiliations:** 1Department of Environmental Medical Biology, Wonju College of Medicine, Yonsei University, Wonju 26426, Gangwon-do, Korea; pharmacisthabib@yonsei.ac.kr (M.H.R.); johnybajgai@yonsei.ac.kr (J.B.); subhamsharma047@gmail.com (S.S.); tththuy@hpmu.edu.vn (T.T.T.); joj2337@naver.com (Y.J.J.); forget419@hanmail.net (S.H.G.); cs-kim@yonsei.ac.kr (C.-S.K.); 2Department of Global Medical Science, Yonsei University Graduate School, Wonju 26426, Gangwon-do, Korea; rokeyahabib94@gmail.com; 3Department of Laboratory Medicine, Yonsei University Wonju College of Medicine, Yonsei University, Wonju 26426, Gangwon-do, Korea; ailynfadriquela@gmail.com

**Keywords:** neurodegenerative diseases, natural products, alzheimer’s disease, parkinson’s disease, therapeutic potential, oxidative stress, neuroinflammation

## Abstract

Natural products derived from plants, as well as their bioactive compounds, have been extensively studied in recent years for their therapeutic potential in a variety of neurodegenerative diseases (NDs), including Alzheimer’s (AD), Huntington’s (HD), and Parkinson’s (PD) disease. These diseases are characterized by progressive dysfunction and loss of neuronal structure and function. There has been little progress in designing efficient treatments, despite impressive breakthroughs in our understanding of NDs. In the prevention and therapy of NDs, the use of natural products may provide great potential opportunities; however, many clinical issues have emerged regarding their use, primarily based on the lack of scientific support or proof of their effectiveness and patient safety. Since neurodegeneration is associated with a myriad of pathological processes, targeting multi-mechanisms of action and neuroprotection approaches that include preventing cell death and restoring the function of damaged neurons should be employed. In the treatment of NDs, including AD and PD, natural products have emerged as potential neuroprotective agents. This current review will highlight the therapeutic potential of numerous natural products and their bioactive compounds thatexert neuroprotective effects on the pathologies of NDs.

## 1. Introduction

A variety of chronic progressive central nervous system disorders triggered by deterioration and eventual loss of neurons are implicated in neurodegenerative diseases (NDs) [1]. Recently, aging of the population has contributed to the increase in NDs [1,2,3], and age-related diseases including NDs are becoming extremely important due to their irreversibility, lack of effective treatment, and accompanied social and economic burdens [4]. Parkinson’s disease (PD), well-characterized by loss of dopaminergic nigrostriatal neurons; Huntington’s disease (HD), which causes loss of spiny, medium-sized striatal neurons; and Alzheimer’s disease (AD) induced by diffuse brain atrophy, are generally known as NDs. Some disorders were often referred to as NDs, including primary dystonia or tremor [5,6]. Patients with NDs manifest with a wide variety of symptoms that often overlap and range from memory and cognitive impairment to impairment of the person’s ability to walk, communicate, and breathe; these patients often have certain clinical characteristics, such as gradual progression over the years, even reaching decades [4]. Furthermore, oxidative stress, neuroinflammation, dysfunction in mitochondria, dysfunctional protein misfolding and agglomeration, and other biological processes have been linked to neurodegeneration [4,7,8]. These biological pathways have been implicated in the development of NDs and their pathogenesis. To date, extensive research has attempted to explain the process and potential therapeutic goals in the battle against NDs [9]. Neuroprotection strategies and relative mechanisms, therefore, function best by interaction with the pathophysiological transition process to interrupt or postpone the neurodegeneration process [4,10,11]. Natural products are known and have been used for their medicinal properties since ancient times. Natural products and their bioactive compounds have been extensively researched and analyzed in recent years, with a focus on biological processes, nutritional principles, potential health, and therapeutic benefits. In recent decades, numerous studies have confirmed the protective effects of natural products and their bioactive compounds against a variety of diseases, including cardiovascular diseases, diabetes, reproductive diseases, cancers, and NDs [12,13,14]. Natural products for the treatment of NDs have emerged as potential neuroprotective agents. This current review highlights the therapeutic potential of numerous natural products and their bioactive compounds that exert neuroprotective effects on the pathologies of NDs.

## 2. Potential Therapeutic Targets of Natural Products against Neurodegenerative Diseases

The mechanism of neuronal damage and death has been investigated for several years, from the organ level to the molecular level. Neurotransmitter accumulation in the brain tissue, particularly glutamate, often leads to excessive brain injury, which can overstimulate nerves and cause neuronal death [15]. According to the World Health Organization figures from 2012, more than 35.6 million people suffer from dementia worldwide, with AD accounting for 60–70% of this population [16]. So far, the pathogenesis of AD has not been fully elucidated. The late onset of sporadic AD, the most prevalent type of the disease, is responsible for genetic vulnerability and environmental factors [17]. To summarize, natural products have recently received increasing interest as alternative or integrative treatment agents against AD and other NDs [18,19]. PD’s neuropathological characteristic consists mainly of the accumulation of intracellular protein aggregates, Lewy bodies, and Lewy neuritis, consisting mainly of the mistreated and aggregated forms of alpha-synuclein protein and the gradual loss of nigrostriatal neurons [20,21]. Mutations in the gene coding for the copper/zinc superoxide dismutase-1 (SOD1) enzyme are linked to amyotrophic lateral sclerosis [22,23]. In addition, HD is a hereditary autosomal dominant neurodegenerative condition marked by adult-onset motor dysfunctions, mental disorders, and cognitive loss [24,25,26]. Moreover, HD is associated with an unstable cytosine–adenine–guanine (CAG) expansion in the huntingtin gene on chromosome 4 [27]. Different biological processes, including oxidative stress and neuroinflammatory and mitochondrial dysfunctions, have been involved in the development and pathogenesis of NDs (Figure 1) [28,29]. Oxidative stress has been emphasized in the progression of AD, PD, and other NDs. In addition, oxidative stress leading to free radical attack on neural cells plays a role in calamitous neurodegeneration [1,30]. However, oxidative stress is caused by an imbalance in the formation of reactive oxygen species (ROS) and a lack of antioxidant defense capacity, resulting in cellular damage, DNA repair system impairment, and mitochondrial dysfunction [10,31]. Oxidative stress also aggravates amyloid-beta (Aβ) generation and aggregation and promotes tau protein phosphorylation that can cause a vicious pathogenic cycle for AD [32,33]. Neuroinflammatory pathways include both the innate and the adaptive immune systems of the central nervous system in connection with neurodegeneration. Furthermore, the pathophysiology of NDs can also include neuroinflammation [34,35]. The main component of the innate immune response is the microglia in the central nervous system. Microglia cause morphological changes in response to pathological changes in the nervous system, and activated microglia secrete a variety of inflammatory mediators including cytokines, chemokines, and cytotoxic molecules. These inflammatory mediators allow astrocytes to respond to the reparation and survival of the secondary inflammatory or growth factor repair response [36,37,38]. Mitochondria are the place of oxidative phosphorylation that help to maintain low cytosol Ca^2+^ concentration [39]. Excessive Ca^2+^ absorption and ROS development lead to a decline inmitochondrial membrane functionand the opening of mitochondrial pores [40]. Several environmental toxins are identified as complex I inhibitors and cause ND-related characteristics [41,42]. The direct association between mitochondrial dysfunction and PD [43,44] was deduced from a discovery of complex I deficiency in the substantia nigra of patients who had died with PD [43,44], followed by evidence of mitochondrial defects in skeletal muscles, platelets, and lymphoblasts in a proportion of cases [45]. The mitochondrial deficiency within the brain appeared to be confined to the substantia nigra. These mitochondrial functional changes occur early prior to the death of the neuron. In the caspase-independent process, the apoptotic factor is converted into the nucleus and results in fragmented DNA or chromatin condensation [46,47]. As neurodegeneration is associated with multifactorial pathological mechanisms, multiple action mechanisms are a promising strategy in ND prevention and therapy.

## 3. Neuroprotective Activities of Numerous Natural Products

A number of natural products have been suggested by Srivastava et al. as traditional pharmacological agents for the treatment of NDs [48]. The use of natural products for the treatment of NDs is widely reported in the literature, as they show different neuroprotective activities. Figure 2 summarizes a wide range of possible therapeutic effects of various natural products for combating NDs.

### 3.1. Luteolin

Luteolin (Lu) is a crystalline yellow flavonoid, common in the plant families Bryophyta, Pteridophyta, Pinophyta, and Magnoliophyta. The food sources of Lu are carrot, onion, celery, olive oil, peppermint, thyme, and oregano [49]. Some Lu molecules have a range of pharmacological properties, including antioxidant, anti-inflammatory, anti-microbial, anti-cancer, and neuroprotective properties [50,51]. These various pharmacological and antioxidant effects are combined with its ability to scavenge oxygen and nitrogen species [52]. A study showed that Lu (20–100 μM) effectively attenuated zinc-induced tau hyperphosphorylation not only through its antioxidant activity, but also through the regulatory mechanisms of the tau phosphatase/kinase system [48]. The decrease in intracellular ROS production increased SOD activity, and the restoration of mitochondrial membrane permeabilization has inhibited caspase-based apoptosis [53]. In addition, the amyloid precursor protein (APP) expression was down-regulated and decreased the secretion of Aβ [54]. In addition, Lu enhanced the nuclear factor erythroid 2-related factor 2 (Nrf2) route and induced the activation of the neuronal cell extracellular signal-regulated kinase (ERK1/2) [55,56]. One of the study reported the concentration of Lu (10–20 μM) increased the neuronal survival, which acts with greater efficacy and equal potency than vitamin E [49].

### 3.2. Quercetin

Quercetin (QCT) is known as a flavonoid in a wide range of food products, such as capers, apples, tomatoes, pasta, green tea, and black and red wines [48]. QCT is a potent herbal antioxidant and is one of the most common flavonoids in edible plants [57]. One study reported the therapeutic efficacy of QCT in improving learning, memory, and cognitive functions in AD [58]. Pharmacologically, QCT has anti-cancer, anti-viral, anti-inflammatory, and anti-amyloid effects [48]. QCT has been described to induce the gradual removal of end products associated with plasma with a recorded half-life of 11–28 h, enabling the body to generate QCT daily [59]. The risk of neurotoxicity can increase through a rise in the number of QCT aglycons entering the central nervous system parenchyma in liposome preparations or by allowing higher blood–brain barrier (BBB) permeability [60]. QCT has been reported to act as a memory booster in a Zebrafish model with scopolamine-induced impairment of memory, potentially improving cholinergic neurotransmission [57]. Further toxicological studies are, therefore, necessary to investigate the risk/beneficial effects of natural products such as QCT [61]. Aβ-mediated apoptosis in hippocampal cultures was significantly reduced at lower doses of QCT (5–20 μM); however, cytotoxicity was induced at high doses (40 μM) [62]. QCT (10 μM) demonstrated anti-amyloidine effects by inhibiting the formation of Aβ fibril [57]. The ability of QCT to cross the BBB and the quantities of QCT and its metabolites in the brain tissue are crucial considerations for its possible in vivo application. QCT reaches the brain, according to in vitro experiments using BBB models [63,64]. Furthermore, QCT and alpha-tocopherol coadministration has been found to promote QCT transport across the BBB [65]. Dihydroquercetin, also known as taxifolin, is a flavonoid commonly found in onions [66]. Treatment with taxifolin prevented spatial memory defects caused by oligomeric Aβ in the wild-type mice hippocampus [67]. In taxifolin-treated cerebral amyloid angiopathy mice, higher blood Aβ levels have been detected, suggesting that Aβ clearance from the brain to bloodwas made easier [68].

### 3.3. Resveratrol

Resveratrol (RSV) belongs toa class of polyphenolic stilbene compounds [69]. RSV is one of the most important red wine flavonoids in grapes, nuts, and other fruits [70]. About 12.5% of participants experienced headaches in the short dose study of RSV, but showed no serious adverse effects [71]. Many studies have reported cardiovascular, anti-cancer, anti-viral, blood-glucose-decreasing, and side effects of RSV [72,73,74]. RSV (10 and 20 mg/kg) primarily works by scavenging ROS as a strong antioxidant by enhancing glutathione (GSH) [75]. The loaded lipid core RSV nanocapsules are elevated compared with free RSV in brain tissue [76]. The gastrointestinal lumen absorbs RSV well, but due to its rapid metabolism and removal, it has poor bioaccessibility [77]. In different forms, the binding of RSV (50 μM) to Aβ was greater, but it was more strongly attached to monomeric Aβ 1–40 than to its fibrillary form [78]. By induction of non-amyloidogenic APP cleavage, RSV reduced Aβ and increased the clearance of Aβ [79]. RSV (100 and 200 μM) can also inhibit C-reactive protein and ERK1/2 mitogen-activated protein kinase (MAPK)[80]. RSV (2.5–40 mg/mL) inhibited the inflammatory response to lipopolysaccharide by reducing inflammatory factors, such as nitric oxide, tumor necrosis factor-α (TNF-α), interleukin (IL)-1β, and IL-6 of astrocytes [81]. Nuclear factor-kappa B (NF-κB) elimination led to a decreased downstream TNF-α and IL-6 levels [82]. A meta-analysis showed that RSV significantly decreased Profile of Mood States (POMS) including vigor and fatigue but had no significant effect on memory and cognitive performance [83]. However, other studies have shown that the BBB plays an important role in Aβ clearance and that its breakdown can result in ineffective clearance [84]. RSV increased claudin-5 expression and decreased the receptor for advanced glycation end products in vivo [85], protecting the BBB integrity [86].

### 3.4. Apigenin

Apigenin (AP) belongs to a subgroup of flavonoids, flavones, based on a skeleton of 2-phenylchromen-4-one (2-phenyl-1-benzopyran-4-one) [87]. To date, very little evidence suggests that AP in a normal diet promotes in vivo adverse metabolic reactions. AP has anti-inflammatory, antioxidant, and anti-cancer characteristics [88]. It is also a strong inhibitor of the enzyme metabolizing several prescription drugs in the body, cytochrome P450 [89]. AP is a highly soluble and intestinally permeable flavonoid. Different transport processes in the intestine can well absorb AP; however, the duodenum is the main absorption site [90]. It also functions as a cell growth, anti-carcinogenic, and enzyme inhibitor, as well as antigenotoxic, anti-inflammatory, and free radical scavenging [91]. In addition, in a double transgenic mouse model of AD (APP/PS1), a review of the neuroprotective potential of AP suggested that apigenin could enhance AD-associated memory impairment, decrease the load of Aβ plaque, and inhibit oxidative stress [92].

### 3.5. Genistein

Genistein is one of the most commonly known isoflavone found in numerous soy products and has been investigated for its antioxidant, anti-inflammatory, and proapoptotic properties; estrogen receptor affinity; protein tyrosine kinase (PTK) inhibition; and other cellular and physiological functions [93]. Current evidence strongly indicates that soy isoflavones protect against a variety of chronic conditions including atherosclerosis, postmenopausal estrogen deficiency, and hormonally based breast or prostate cancer [94]. Recently, some researchers have found the neuroprotective activity of genistein. A study reported that genistein (100 μM) was found to be effective against toxicity induced by the Aβ31-35 peptide in primary neuronal cells obtained from newborn Wistar rats [95]. It has been confirmed that genistein, a phytoestrogen able to cross the BBB, has antioxidants from ultraviolet light and chemical insults. Another research reported that in cultured hippocampal neurons, genistein has a neuroprotective effect against Aβ25-35-induced apoptosis [96]. Exposure to aged Aβ25-35 for 24 h has been shown to double the DCF fluorescence strength compared with controls for 24 h [97]. Emerging evidence indicates that estrogen and estrogen-like chemicals have beneficial effects on ND, especially PD. Interestingly, genistein exhibited a preventive effect on neuronal degeneration caused by increased oxidative stress [98]. In addition, genistein can cross the BBB [99], and it has proven to be safe for a long time (over 1 year) in the clinical trial at concentrations up to 150 mg/kg/day.

### 3.6. Hesperidin

Flavanone-glycosides rich in citrus fruit, lemon, sweet orange, and grapes are also called hesperidin (C_28_H_34_O_15_) [100]. Hesperidin administration for 16 wks helped boost learning and memory function by increasing the recognition index in the transgenic mouse model of APPswe/PS1dEE [101]. It corrects mitochondrial disorders caused by Aβ by lowering levels of malondialdehyde and hydrogen peroxide and restoring GSH depletion and total antioxidant ability (T-AOC). A protein kinase that has a prominent role in mitochondria and AD functions is glycogen synthase kinase-3β (GSK-3β). It has an important impact on the protein tau hyperphosphorylation and the mitochondrial target [102]. Increasing oxidative damage triggers the activation of this protein kinase. By inhibiting the restoration of this kinase, hesperidin theoretically rescued cognitive deficits and showed mitochondrial neuroprotective effects. It was the potential mechanism by which hesperidin lowered the Aβ1-40 level [100]. Hesperidin also inhibited learning and memory impairments resulting from aluminum chloride (AlCl_3_)-induced AD, functioning as an acetylcholinesterase inhibitor. In the rat hippocampus and brain cortex, hesperidin attenuated APP expression through the NF-κB-dependent pathway and suppressed Aβ1-40 and β-and γ-secretase levels [49,103]. The neuroprotective role of hesperidin was reported in the signals of up-regulating B-cell lymphoma 2 (Bcl2) and down-regulating Bcl-2-associated X protein (Bax) [104,105,106]. In addition, hesperidin has been reported to have neuroprotective effects in many neurological disorders, such as cerebral ischemia, HD, and PD, at 50 and 100 mg/kg oral doses [107]. The hesperidin of citrus flavonoid has neuroprotective effects andmay pass through the BBB. Hesperidin inhibits the release of glutamate and exercises an excitotoxic neuroprotection in rat hippocampus with kainic acid [108].

### 3.7. Uncaria Rhynchophylla


The herb *Uncaria rhynchophylla*, part of the Rubiaceae family, is used in traditional Chinese medicine. *Uncaria rhynchophylla* extract is made up of alkaloids, rhinchophylline, hirsutine, hirsuteine, corynanthine, corynoxine, and dihydrocorynantheine [109,110]. The most widely studied and named neuroprotective compositions among the alkaloids are rhinchophylline and isorhynchophylline [111,112]. In addition, the neuroprotection effect of *Uncaria rhynchophylla* has been reported in an experimental PD model [113]. Shim et al. documented that *Uncaria rhynchophylla* reduced neuronal cell death and ROS production, restored GSH levels in PC12 cells in case of toxicity of caspase-3 and 6-hydroxydopamine (6-OHDA) cells, and reduced the neuronal loss in the substantia nigra dopaminergic rats induced by 6-OHDA [113,114]. In the 1-methyl-4-phenylpyridinium (MPP^+^) induced SH-SY5Y and 1-methyl-4-phenyl-1, 2, 3, 6-tetrahydropyridine (MPTP) mouse cell models, *Uncaria rhynchophylla* has been found to improve cell viability, attenuate dopaminergic neuronal significant nigra and striatum lowering, and inhibit heat-shock protein 90 and autophagy [115]. All these results, taken together, indicate that *Uncaria rhynchophylla* demonstrates neuroprotective activity through multiple mechanisms of neuronal defense against damage, which may be attributable to the beneficial combined action of the active compounds in *Uncaria rhynchophylla*. These antioxidant compounds inhibit the anti-inflammatory effect by inflammatory mediation and the anti-apoptotic effect, modulating the event and preventing the activation of the caspase and decreasing ROS generation and improving the antioxidant protective mechanism. Rhynchophylline is an alkaloid found in certain *Uncaria* species (Rubiaceae). However, recent studies revealed that isorhynchophylline can easily pass the BBB. These observations suggested that isorhynchophylline may be an anti-inflammatory substance used to treat NDs [116,117].

### 3.8. Marine Macroalgae


Marine macroalgae are plant-like organisms, typically referred to as seaweed, that commonly live in coastal areas. The three groups can be categorized as brown (Phaeophyceae), red (Rhodophyceae), and green algae (Chlorophyceae) [118]. Phenolic compounds, proteins, peptides, pigments, amino acids, and phenols are also found in a variety of bioactive materials [119]. Numerous studies find that algae and bioactive compounds of various algae have a health impact [118,120,121]. In addition, Pangestuti et al. showed that carotenoids have a high radical scavenging function and are present in marine algae as a major antioxidant [122]. Furthermore, another study found that marine extracts increase cell viability, decrease oxidative stress, have a healthy mitochondrial membrane potential, and decrease caspase-3 activities. This indicates the neuroprotective effects and the antioxidant properties of these algae [123]. Silva et al. suggested the possibility of mediating this neuroprotective action with antioxidant compounds in algae extracts [123]. However, the researchers’ concern about the potential use for pharmaceuticals, particularly when new drug delivery systems are being developed, recently attracted their attention to marine sulfated polysaccharides [124,125]. The biological activities of sulfated polysaccharides have been identified in various studies [126,127]. In the meantime, *Undaria pinnatifida* fucoidan improved cell viability, prevented apoptosis via inhibition of activation of caspase-3, and enhanced dense antioxidant systems in Aβ (25–35), SOD activity, and GSH materials in PC12 cells with neurotoxicity [128]. The fucoidans have a reduced aggregation of Aβ (1–42), decreased cytotoxicity (1–42), and PC12 hydrogen peroxide caused by Aβ, decreased Aβ-induced apoptosis (1–42), and improved the role of neuritis outgrowth [129,130]. Moreover, the possibility of developing marine algae components as neuroprotective agents has not been investigated because of the BBB.

### 3.9. Cyanobacteria


Cyanobacteria are prokaryotic, photosynthetic, self-producing species that are closely related to bacteria and are commonly referred to as blue-green algae. They are members of the Oscillatoriaceae family. Researchers have been very attentive to their potential pharmacological properties and advantages for various medical conditions [120,131]. *Spirulina platensis* is a multicellular planktonic, alkaliphilic cyanobacterium. It has been widely studied and recognized for its proper nutritional components. Subsequently, it may protect itself against dopaminergic neuronal loss triggered by MPTP in substantia nigra. *Spirulina platensis* has anti-inflammatory and antioxidant properties that help it defend against PD caused by 6-OHDA [132,133]. However, evidence suggests that polysaccharides derived from *Spirulina platensis* have an antioxidant effect on dopaminergic neurons and dopamine levels, rather than inhibition of monoamine oxidase B [134]. These findings showed that *Spirulina maxima* extract improved cognitive impairment by inhibiting Aβ accumulation [135]. In addition, the neuroprotective role of *Spirulina maxima* (Sp.) against MPTP neurotoxicity, used as a model of PD [122]. Other studies have shown that *Spirulina maxima* extract has protected against memory damage caused by scopolamine in mice [136,137]. These results show that, via antioxidant activity, *Spirulina maxima* exert its neuroprotective impact [136]. In addition, oral administration of c-phycocyanin, a component of Spirulina, has an effect in the hippocampus, because it crosses the BBB [138]. These studies have shown collectively that cytoprotective activity against neurodegeneration is demonstrated by different mechanisms of action, primarily by antioxidants.

## 4. Role of Other Natural Products in Neurodegenerative Diseases


NDs exhibit some common characteristics despite specific clinical and etiopathogenic differences, such as irregular protein deposition, abnormal cellular transports, mitochondrial deficits, inflammation, intracellular Ca^2+^ overload, unregulated ROS generation, and excitotoxicity [4,139]. In the pathogenesis of all essential NDs, reactive astroglia and/or microglia are also involved [140,141]. Several natural substances have been suggested for treating NDs to complete and/or help conventional pharmacological agents [4]. Their use on NDs is commonly identified as a consequence of several different neuroprotective activities reported in the literature [142,143,144]. The main objectives include mitochondrial dysfunction, inflammation, oxidative stress, and protein malfunction among the natural products [145,146,147]. Some animal products, such as omega-3 fatty acids, inhibit cell toxicity and have anti-inflammatory effects in the treatment of AD [148]. Plant-based products, such as lunasin, polyphenols, alkaloids, and tannins, are possible therapeutic candidates for AD [149]. Resveratrol and flavonoids appear to be dietary additives that have obvious neuroprotective and other beneficial effects on human cognitive disability [69,150]. Although natural products can be extracted from different biological sources, it is not trivial to turn them into therapies. The challenges can include concerns about their stability and neuro-availability, difficulties in properly defining and quantifying the active principle, and, lastly, difficulties in organizing large-scale clinical trials to evaluate these complex products [151]. The capacity to defend against neurodegeneration has been evaluated in several differentnatural products. Table 1 and Table 2 provide a description of natural products and their bioactive compounds with various neuroprotective functions, depending on the disease being treated. Natural products and their bioactive substances with neuroprotective function in the treatment of AD are represented in Table 1. Similarly, PD treatment currently includes medicines such as Levodopa, primarily catalytically converted into dopamine by dopa decarboxylase in the brain, resulting in its therapeutic effects [152,153]. There is evidence that correlates neuronal mitochondrial dysfunction with the pathogenesis of PD [154,155]. However, this dysfunction is associated with the abnormal accumulation of α-synuclein, which causes an alteration of normal mitochondrial function, leading to neuronal degeneration and strong oxidative stress [156,157]. In addition, the presence of neuroinflammation is another peculiar characteristic of PD, which plays a significant role in the development of the disease. However, the inflammation depends also on the impaired energy metabolism at the level of the mitochondria impairment that causes the activation of the microglia and the relative generation of a plethora of pro-inflammatory mediators, including prostaglandins, cytokines, chemokines, complement, proteinases, ROS, and RNS [158]. Moreover, most patients with PD also have non-motor symptoms, including disorders of the sleep–wake cycle regulation, cognitive impairment disorders of mood and affect, autonomic dysfunction, as well as sensory impairmentand pain. Recently, the management of age-related diseases, such as PD, has been associated with consumption of functional food or food supplements. Certainly, a healthy diet rich in foods containing antioxidants, vitamins, and minerals or the use of food supplements can help to reduce the symptoms of PD and the related pathological mechanisms [158]. *Mucunapruriens* belongs to the family Leguminosae and is a twiner with trifoliate leaves, purple flowers, and pods covered with hairs. Seeds from *Mucuna pruriens* (Atmagupta) have been described as a useful therapeutic agent in different diseases of the human nervous and reproductive system, including PD in the ancient Indian medical system, Ayurveda [158]. *Mucuna pruriens* exhibited twice the anti-parkinsonian activity compared with synthetic levodopa, suggesting that *Mucuna pruriens* may contain unidentified antiparkinsonian compounds in addition to levodopa, or that it may have adjuvants that enhance the efficacy of levodopa [159]. Another therapy involves anticholinergic drugs that can block the excitability of cholinergic nerves by striatal cholinergic receptors; it has also been shown that they can suppress dopamine reuptake to increase the activity of dopaminergic neurons [160]. Natural products and their bioactive substances with neuroprotective function in the treatment of PD are shown in Table 2. The therapeutic potential of medicinal plants has been studied and evaluated in scientific circles. Numerous medicinal plants extract used in the clinical trial and their outcomes are shown in Table 3. In conclusion, as complementary or integrative therapeutic agents against AD, PD, and other NDs, natural products have recently gained greater attention [161].

## 5. Limitations, Future Prospects, and Challenges


The capacity of neuroprotection and the development of therapeutic products and tools, including isolated natural compounds, against various NDs have been naturally developing. Despite the promising neuroprotective activity in pre-clinical settings, the translation of promising preclinical investigations to clinical use has proven difficult because human clinical studies of neurodegenerative disorders have no favorable findings. Natural products and isolated natural compounds face several challenges and weaknesses that can compromise their therapeutic efficacy, including poor bioavailability and decreased water solubility, physical and chemical instabilities, rapid metabolism, and BBB crossing. These reviews of the literature provide more details [199,200,201]. However, numerous natural compounds, including resveratrol [202] and curcumin [203,204], have been reported to have low bioavailability and limited stability due to degradation or transformation into inactive derivatives [205,206]. As a result, their efficacy is reduced. In addition, the BBB prevents access of natural compounds to the brain, thus prevents them to reach their action site. This limits their distribution to the brain tissue and results in low bioavailability [207]. Nanotechnology and nanocarriers can help improve therapeutic responses and effectiveness in the delivery of natural products and their isolated compounds, which will help solve these problems [208,209]. Nanoparticles may be used in the delivery system to increase the bioavailability of natural products and their compounds. Polymeric nanoparticles, nanogels, rigid lipid nanoparticles, crystalline nanoparticles, micelles, and dendrimer complexes are the most commonly used nanoparticles [210,211]. Several studies have been published on the use of natural nanoparticles with thesecompounds, such as epigallocatechin 3-gallate for treating AD [212], rosemary acid for HD [213], curcumin for brain disease [214].

## 6. Concluding Remarks


Therapeutic potential for natural products and natural bioactive compounds to be neuroprotective has been supported by various research studies. Natural products and important bioactive compounds are needed to prevent and treat various NDs without causing harmful adverse effects. Since several functional pathways are found in neurodegenerative pathologies, ND prevention and treatment approaches have an important role to play. For natural products and bioactive substances, it is preferable to use various modes of action to display neuroprotective effects. Furthermore, the ability of natural products and their bioactive compounds to cross the BBB is essential for neuroprotective activity. It is important to develop new methods and techniques, such as nanotechnology, for the delivery of natural compounds and drugs in order to enhance the role of natural products and bioactive compounds in ND prevention and therapeutic fields, in order to promote access to the brain of neuroprotective products.

## Figures and Tables

**Figure 1 molecules-26-05327-f001:**
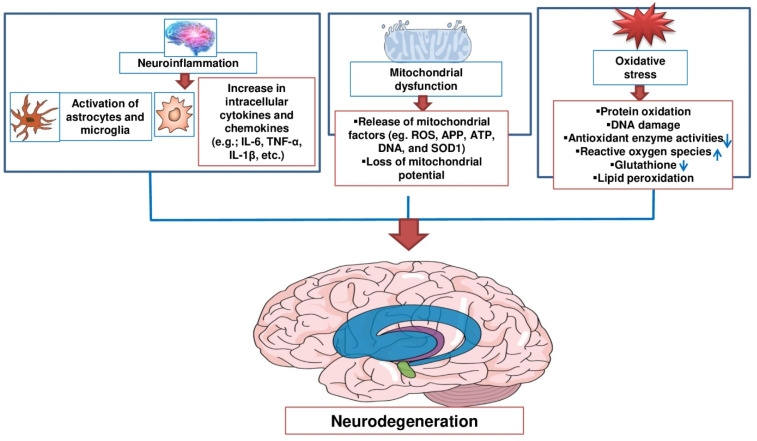
Different biological processes, including oxidative stress and neuroinflammatory and mitochondrial dysfunctions, have been involved in the development and pathogenesis of NDs.

**Figure 2 molecules-26-05327-f002:**
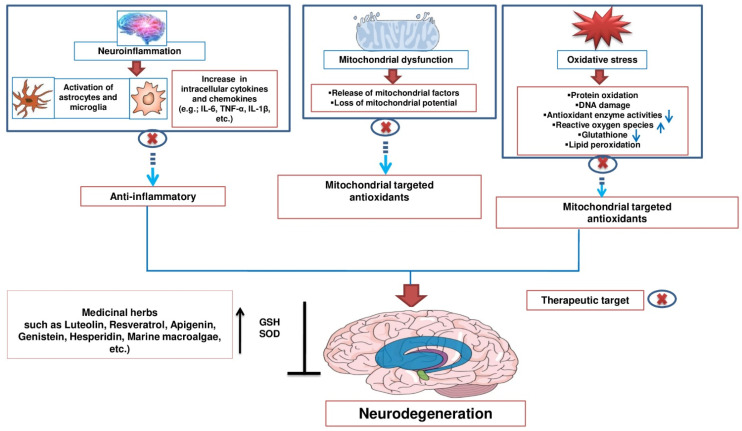
Therapeutic effects of numerous natural products for combating NDs.

**Table 1 molecules-26-05327-t001:** Representative natural products and their bioactive substances with neuroprotective activity in the treatment of AD related disease model.

Name of Plant Part	Name of Model	Neuroprotective Mechanisms	References
Yacon (Poepp. and endl.) (*Smallanthus sonchifolius*) extract of the leaf	Rat	Memory deficits prevented	[162]
Natural safflower aqueous extract	Rat	Short and long-term memory improved	[163]
Methanolic extract of *Lactucacapensis* thunb. leaves	Rat	Lowering the degree of lipid peroxidation and protein oxidation	[164]
Turmeric powder	Human	Improvement in the quality of life and behavioral symptoms	[165]
*Tabernaemontana divaricata * root extract	Mouse	Prevented memory loss	[166]
Coconut oil enriched Mediterranean diet	Human	Enhanced cognitive features	[167]
Osmotin, a protein derived from *Nicotiana tabacum*	Mouse	Increased conduct of random alternation	[168]
Germinated brown rice	SH-SY5Y cells	Reduced production of intracellular ROS	[169]
Isolated from *Huperzia serrata* is Huperzine A	Human	Improvement in functions of memory, cognition, and actions	[170]
Huperzine A isolated from *Huperzia serrata*	Rat	Reduce oxidative damage	[171]

**Table 2 molecules-26-05327-t002:** Representative natural products and their bioactive substances with neuroprotective activity in the treatment of PD related disease model.

Name of Plant Part	Name of Model	Neuroprotective Mechanisms	References
Smith ethyl acetate extract *Zingiber zerumbet* (L.)	Rat	Prevention of neuronal damage	[172]
*Urticadioica * Linn. ethyl acetate fraction.	Rat	Enhanced motor control and alteration in oxidative protection	[173]
Apium *graveolens* L.	Mouse	Improved behavioral disorder caused by MPTP	[174]
*Tribulus terrestris * extract	Mouse	Improved the proportion of viable neurons	[175]
Ethanol extract of *Tinospora cordifolia*	Rat	Restored locomotive operation behavioral changes caused by 6-OHDA	[176]
Dihydromyricetin (DHM) (*Ampelopsis grossedentata*)	Mouse	Mitigated the deficit in the balance of movement caused by the MPTP	[177]
Agaropentaose, an agaro-oligosaccharide monomer that is isolated from red algae hydrolysates of agarose	SH-SY5Y cells	Inhibited potential loss of mitochondrial membrane	[178]
*Capsicum annuum * L. extract	Mouse	Restored development of cholinesterase in the brain	[179]
β-Caryophyllene, a cannabinoid compound originating from a plant known as phytocannabinoids	Rat	Lipid peroxidation inhibited	[180]
Viride var. of Coeloglossum. Extract from bracteatum	Mouse	Prevented neuronal dopaminergic loss	[181]
Boswellic acids	Rat	Motor functions improved	[182]
Rosmarinic acid isolated from callus of *Perilla frutescens*	Rat	Increased tyrosine hydroxylase numbers	[183]
Olive leaf extracts (*Olea europaea* L.)	Rat	Inhibited tyrosine hydroxylase-positive neuron depletion	[184]
*Oxalis corniculata * extract	Mouse	Improved preservation and retrieval of memory	[185]
Curcuminoids (*Curcuma longa* (L.) rhizomes)	Mouse	In the striatum, decreased proinflammatory cytokine and complete nitrite production	[186]
Supplementation of fish oil (rich in omega-3 polyunsaturated fatty acids)	Rat	Reduced loss of substantia nigra neurons and nerve terminals in the striatum)	[187]
Germinated brown rice	Rat	Improved the number of dopaminergic neurons that survive	[188]

**Table 3 molecules-26-05327-t003:** Numerous medicinal plant extracts used in clinical trials and their outcomes.

Plant Species	Type of Clinical Study	Clinical Outcomes	Reference
*Salvia officinalis*	Randomized, double-blind	Significantly improved cognitive function	[189]
Resveratrol	Randomized, placebo-controlled, double-blind, multicenter 52-wk phase 2 trial	Resveratrol was safe and well-tolerated. Resveratrol and its major metabolites penetrated the blood–brain barrier to have CNS effects	[190]
*Ginkgo biloba * L.	Longitudinal, 3 monthly follow-ups over a 12-month period	Focal electroretinograph↑ amplitude and sensitivity amplitude that stabilized after 3 months independent of genotype	[191]
*Crocus sativus * L.	Longitudinal, open-label study, 8 monthly follow-ups over a 29 (±5)-month period	Focal electroretinograph saffron treated age-related macular degeneration patients: Visual function remained stable	[192]
*Curcumin longa*	24 older adults with physical or cognitive impairment	Improve physical function and cognitive function	[193]
*Crocus sativus*	Depressant patients	The effect of *C. Sativus* similar to imipramine in the treatment of mild to moderate depression	[194]
*Nigella sativa*	Asthmaticpatients	Improvement of all asthmatic symptoms, chest wheeze and pulmonary function test values	[195]
*Centella asiatica*	Randomized, double-blind placebo-controlled trial	Improved memory function	[196]
*Bacopa monnieri*	Double-blind, placebo-controlled trial in 38 healthy volunteers (aged 18–60 years)	Significantly improved cognitive function	[197]
*Withania somnifera*	Prospective, randomized, double-blind, placebo-controlled	Significantly improved executive functions in adults with mild cognitive impairment	[198]

## Data Availability

The data presented in this study are available within the article (tables and figures).

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
