# Peer review of "Therapeutic Potential of Natural Products in Treating Neurodegenerative Disorders and Their Future Prospects and Challenges"

_molecules, 2021, doi:10.3390/molecules26175327_

Round 1
Reviewer 1 Report
Well done, it is a nice review
Author Response
Thank you for these comments and for the careful reviews that you have given for our manuscript. Authors really appreciate for your comments regarding our manuscript.
Reviewer 2 Report
The paper presents analysis of current space for natural medicine in neurodegenerative diseases and complements some of the recently published reviews in this space (e.g. Ciulla et al, 2019 - PD and nutrition). Some specific feedback:
- consider removing Figure 2 - as the current treatments are not really discussed this figure is not adding any further information and is mentioned only once without further explanation
- consider incorporating some of the other herbal medicines as part of the PD section (e.g. Mucuna pruriens and PQQ (high in number of foods)) - as the review is not systematic in nature it did miss some of the literature connected to these diseases
- consider summarising the actions of the mentioned compounds within each section - it currently reads like a list of papers with no real summary of the actions
- structure of the literature and English needs to be revised as it deters from the message the review wants to convey
Author Response
Thank you for these comments and for the careful reviews that you have given for our manuscript.

Reviewer 3 Report
The manuscript “Therapeutic potential of natural products in treating neurodegenerative disorders and their future prospects and challenges”. The manuscript is well written; however, it requires few changes that could help to increase the overall quality of the manuscript.
Please include a table to summarize herb/plant extract's role in the clinical trial and their outcome.
Figure 1- Please modify the figure to include more specific mediators of neuroinflammation (include which cytokines/chemokines are predominantly involved). Similarly, the statement release of mitochondrial factors, please mention the specific factors.
Figure 2 – Mention the herbs that were acting as anti-inflammatory and mitochondria-targeted antioxidants.
Line 327 – 328: Please modify the statement – “In MPTP-induced striatum mice”
Author Response

(The authors gave the same response as above.)
